

# Targeting fibroblast growth factor receptors causes severe craniofacial malformations in zebrafish larvae

Liesbeth Gebuijs[1,2,3], Frank A. Wagener[1,2], Jan Zethof[3], Carine E. Carels[4], Johannes W. Von den Hoff[1,2] and Juriaan R. Metz[3]

[1] Department of Orthodontics and Craniofacial Biology, Radboud University Nijmegen Medical Center, Nijmegen, The Netherlands
[2] Radboud Institute for Molecular Life Sciences, Nijmegen, The Netherlands
[3] Department of Animal Ecology and Physiology, Radboud University Nijmegen, Nijmegen, Netherlands
[4] Department of Human Genetics and Department of Oral Health Sciences, KU Leuven, Leuven, Belgium

## ABSTRACT

**Background and Objective**. A key pathway controlling skeletal development is fibroblast growth factor (FGF) and FGF receptor (FGFR) signaling. Major regulatory functions of FGF signaling are chondrogenesis, endochondral and intramembranous bone development. In this study we focus on *fgfr2*, as mutations in this gene are found in patients with craniofacial malformations. The high degree of conservation between FGF signaling of human and zebrafish (*Danio rerio*) tempted us to investigate effects of the mutated *fgfr2*[sa10729] allele in zebrafish on cartilage and bone formation.

**Methods**. We stained cartilage and bone in 5 days post fertilization (dpf) zebrafish larvae and compared mutants with wildtypes. We also determined the expression of genes related to these processes. We further investigated whether pharmacological blocking of all FGFRs with the inhibitor BGJ398, during 0–12 and 24–36 h post fertilization (hpf), affected craniofacial structure development at 5 dpf.

**Results**. We found only subtle differences in craniofacial morphology between wildtypes and mutants, likely because of receptor redundancy. After exposure to BGJ398, we found dose-dependent cartilage and bone malformations, with more severe defects in fish exposed during 0–12 hpf. These results suggest impairment of cranial neural crest cell survival and/or differentiation by FGFR inhibition. Compensatory reactions by upregulation of *fgfr1a*, *fgfr1b*, *fgfr4*, *sp7* and *dlx2a* were found in the 0–12 hpf group, while in the 24–36 hpf group only upregulation of *fgf3* was found together with downregulation of *fgfr1a* and *fgfr2*.

**Conclusions**. Pharmacological targeting of FGFR1-4 kinase signaling causes severe craniofacial malformations, whereas abrogation of FGFR2 kinase signaling alone does not induce craniofacial skeletal abnormalities. These findings enhance our understanding of the role of FGFRs in the etiology of craniofacial malformations.

Corresponding author
Johannes W. Von den Hoff,
hans.vondenhoff@radboudumc.nl

## INTRODUCTION

The formation of the vertebrate skeleton is tightly controlled by a large number of genes that are part of a network of interconnected signaling pathways. Pathways involved in skeletal development include bone morphogenic protein (BMP), transforming growth factor beta (TGF-$\beta$)-, Hedgehog-, and Wnt/$\beta$-catenin signaling (*Ahi, 2016*; *de Caestecker, 2004*; *Huangfu & Anderson, 2006*). Another pathway crucial in different stages of skeletal development is fibroblast growth factor (FGF) signaling. There are currently 22 FGFs known in humans, and five FGF receptors (FGFRs) are expressed. Four of the five distinct but highly homologous transmembrane receptors (FGFR1-4) consist of a signal peptide, three extracellular immunoglobulin-like domains (Ig1-3), an acid box domain between Ig1 and 2, a transmembrane domain and an intracellular split tyrosine kinase domain (*Hatch, 2010*). The fifth FGFR (FGFR5) lacks the tyrosine kinase domain but has a short intracellular tail instead and signals through dimerization with FGFR1 (*Amann & Trueb, 2013*; *Wiedemann & Trueb, 2000*). Binding of an FGF ligand to an FGFR induces receptor dimerization, which leads to transphosphorylation and subsequent activation of a number of downstream cascades including Ras/MAP kinase, phospholipase C $\gamma$, PI3-kinase and STAT (*Ornitz & Itoh, 2015*; *Ornitz & Marie, 2002*; *Thisse & Thisse, 2005*). These pathways regulate developmental processes like cell proliferation, migration and differentiation, all necessary for bone and cartilage formation (*Ornitz & Itoh, 2015*).

Genes encoding FGFRs have distinct expression patterns during mammalian skeletal development and their main functions are regulation of endochondral and intramembranous bone development, chondrogenesis and bone mechanical sensing (*Du & Carlin, 2012*; *Papachristou et al., 2009*). *FGFR1* and *FGFR2* are expressed in mesenchymal osteoprogenitors and differentiating osteoblasts. Both receptors are crucial for proliferation and maturation of osteoblasts (*Jacob et al., 2006*). When condensed mesenchymal cells begin to differentiate into chondrocytes, *FGFR2* expression decreases while *FGFR3* expression increases (*Peters et al., 1992*). *FGFR3* is induced in proliferating and differentiating chondrocytes that give rise to, among others, the Meckel's cartilage (*Havens et al., 2008*). *FGFR4* is expressed in developing craniofacial muscles, such as the tongue and those that attach to the facial bones (*Rice, Rice & Thesleff, 2003*). In the human craniofacial region, FGF signaling through FGFRs is also crucial for the formation of the facial primordia that give rise to almost all facial structures, including the palate (*Nie, Luukko & Kettunen, 2006*). Messenger RNA of FGFR1-3 can undergo alternative splicing to yield protein variants with specific binding affinity for different FGF ligands (*Zhang et al., 2006*). The splicing of FGFR1 and its ligand binding properties resemble those of FGFR2 and these receptors often show functional redundancy during development (*Leerberg, Hopton & Draper, 2019*; *Ornitz & Itoh, 2015*).

In the current study we first focus on FGFR2 and its role in craniofacial development. Specific mutations in the *FGFR2* gene have been identified in patients with craniofacial malformations, such as cleft of the lip and/or palate (CLP); most of the mutations in *FGFR2* causing syndromes are gain-of-function mutations (*Hajihosseini et al., 2001*; *Snyder-Warwick et al., 2010*). For instance, two closely neighboring mutations in the *FGFR2*

Ig3a domain cause 98% of all Apert syndrome cases (*Wilkie et al., 1995*). On the contrary, *FGFR2* mutations in patients with non-syndromic CLP mainly cause a loss-of-function (*Riley et al., 2007*). The two reported mutations in the Ig1 domain in these patients both influence the autoinhibition state of the receptor, in which a conformational change of the receptor prevents binding of FGFs. Conditional silencing of the exons 8-10 of the *Fgfr2* gene in mice, causing loss of the Ig3 and transmembrane domain, resulted in skeletal dwarfism, a domed-shaped skull and a decrease in bone density (*Yu et al., 2003*). This indicates that multiple domains in FGFR2 are of high functional importance and that the location of the mutation determines the phenotype. More mouse studies revealed that phenotypes caused by mutations in distinct domains in *Fgfr2* result from disturbed osteoblast proliferation and differentiation (*Holmes et al., 2009*; *Liu et al., 2013*; *Wang et al., 2005*; *Yin et al., 2008*). However, null mutations of *Fgfr1* and *Fgfr2* have been found to be embryonically lethal in the mouse, hindering further research (*Arman et al., 1998*; *Deng et al., 1994*; *Yamaguchi et al., 1994*).

The zebrafish (*Danio rerio*) is widely recognized as valuable model system for the study of human genetic diseases (*Lieschke & Currie, 2007*; *Machado & Eames, 2017*; *Phillips & Westerfield, 2014*). Early craniofacial development in zebrafish is to some extent comparable to that in mammals. Indeed, the signaling pathways involved in skeletal development, including sonic hedgehog (SHH), BMP and FGF are highly conserved and the anterior neurocranium of the fish is a genetically tractable model for the mammalian hard palate (*Mork & Crump, 2015*; *Swartz et al., 2011*). In zebrafish, there are 31 FGFs known, because paralogs of some FGFs exist due to the evolutionary genome duplication in teleost fish (*Glasauer & Neuhauss, 2014*). Zebrafish have five widely expressed *fgfr* genes: *fgfr1a*, *fgfr1b*, *fgfr2*, *fgfr3* and *fgfr4* (*Ornitz & Itoh, 2001*).

In the present study, we investigated the effects of the mutated *fgfr2*[sa10729] allele on craniofacial development. This is a point mutation (C >T at position 1261 of the coding sequence) in exon 9, which causes a premature stop codon. The resulting protein is terminated between the transmembrane and intracellular domain of the receptor (Fig. 1). We hypothesize that the lack of the intracellular signaling domain renders the receptor non-functional and affects craniofacial development through changes in the expression of genes involved in cartilage and bone formation. We therefore investigated the development of cartilaginous and mineralized elements in the zebrafish head at 5 dpf in *fgfr2*[sa10729] homozygous (−/−) and heterozygous (+/−) mutant larvae, which were compared to wild types. In addition, the relative expression of *fgfr2* itself and 16 other genes related to bone and cartilage formation was determined. The same analyses were performed after pharmacological blocking of FGFR1-4 with the pan-FGFR kinase inhibitor BGJ398, exposed during 0–12 or 24–36 h post-fertilization (hpf). Those windows were chosen because the 0–12 hpf period corresponds to the formation of the neural tube (*Araya et al., 2016*; *Kimmel et al., 1995*), from which cranial neural crest cells (CNCCs) migrate to the head region (*Kague et al., 2012*; *Schilling & Kimmel, 1994*). On the other hand, during 24–36 hpf the CNCCs differentiate into chondrocytes that contribute to the craniofacial elements in the embryonic head (*Kague et al., 2012*; *Wada et al., 2005*). This experimental

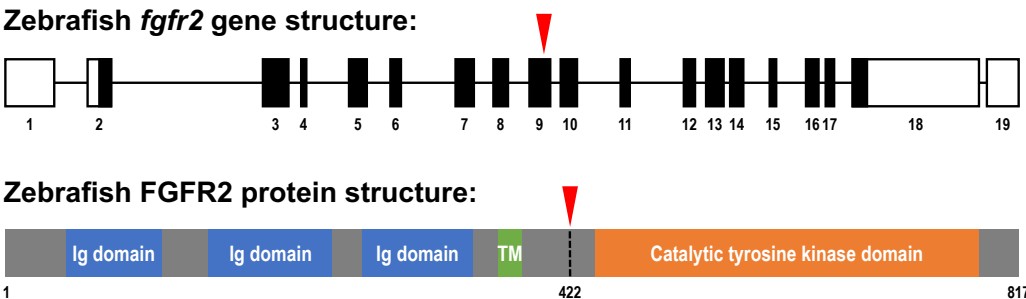

**Figure 1** **Schematic representation of the zebrafish *fgfr2* gene and protein structure.** (A) Five highly similar transcript variants have been found; variant 1 is depicted here (transcript fgfr2-201 in Ensembl; NCBI Reference Sequence: NM_001243004.1). The *fgfr2* gene consists of 19 exons, of which 17 coding exons. Note that the exons (depicted as boxes) are drawn on a 20-times larger scale than the introns (depicted as lines). The full gene transcript is 4,741 bp long; the coding sequence is 2,451 bp running from exon 2 to exon 18 (indicated by black filling). The red arrowhead pointing down indicates the site of the mutation in the *fgfr2*[sa10729] mutant zebrafish used in the current study. (B) The gene encodes an 817 amino acid protein. The protein is characterized by three immunoglobulin (Ig) domains, a transmembrane (TM) region and a tyrosine kinase domain. The truncated FGFR2 protein that results from the mutation consists of 421 amino acids (premature stop codon indicated by the red arrowhead pointing down). (C) Partial nucleotide (lower case) and amino acid (upper case) sequence detailing the point mutation. The c > t mutation causes a stop codon (indicated by the asterisk).

design will enable us to identify the critically sensitive windows for disturbances in FGF signaling.

# MATERIALS & METHODS

## Zebrafish husbandry and breeding

Zebrafish (*Danio rerio*) were raised and kept at the Radboud Zebrafish Facility, at 28 °C and under a 14hr light/10hr dark cycle with feedings twice daily. The mutant line *fgfr2*[sa10729] was obtained from the European Zebrafish Resource Center (EZRC, Karlsruhe Institute of Technology, Germany). To obtain homozygous mutants (−/−), heterozygotes (+/−) and wild type siblings (+/+), adult heterozygous carriers were pair-wise crossed. Genotyping was performed as described below. Every two weeks, breeding tanks were set up the day before mating, with males and females separated by a transparent divider. After turning on the lights at 9 AM, the water was changed for low conductivity, warm water of ∼30 °C and the divider was removed to initiate spawning. Fertilized eggs were collected within 60 min after spawning and transferred to Petri dishes with standard E3 medium (5 mM NaCl, 0.17 mM KCl, 0.33 mM CaCl$_2$, 0.33 mM MgSO$_4$ and 0.00001% methylene blue). Petri dishes with embryos were placed in an incubator set at 28 °C with a 14 h light/10 h dark cycle

to grow up. E3 medium was refreshed at days 1, 4 and 5 and any unfertilized eggs, dead embryos and post-hatching chorions were removed. We complied with all guidelines set in the European Union Directive 2010/63/EU on the protection of animals in research.

## Cartilage and bone staining

At 5 dpf, all larvae were euthanized in 0.1% (v/v) 2-phenoxyethanol and fixated for 30 min in 2% paraformaldehyde (pH = 7.4 with PBS). After a washing step of 10 mins with 100 mM Tris pH7.5/10 mM MgCl$_2$, larvae were stained according to a slightly modified Kimmel Lab protocol (*Walker & Kimmel, 2007*). The only adjustments to this protocol were staining with Alcian Blue for 2 h and staining with Alizarin Red S overnight, with all other solutions as described. For imaging, larvae were transferred from the destaining solution to 100% glycerol.

## Imaging

Imaging was performed as described previously (*Gebuijs et al., 2019*). Stained zebrafish larvae in 100% glycerol were embedded in a round borosilicate glass capillary (CV6084-100; Vitrocom, Mountain Lakes, NJ, USA), which was placed inside a square borosilicate capillary (CV8290-100), also filled with 100% glycerol. The capillaries were placed in a sample holder with an axial rotating system (adapted from *Bruns, Schickinger & Schneckenburger (2015)*) and images were acquired from dorsal, ventral and lateral sides of the larvae under a microscope using Leica Application Suite (LAS 3.3, Leica, Wetzlar, Germany).

## Cartilage and bone analysis

Images were imported in Fiji (*Schindelin et al., 2012*) and the following craniofacial cartilage parameters were measured: (1) total head length, (2) length ceratohyal to anterior end of the head, (3) length ceratohyal to posterior end of the head, (4) width at Meckel's cartilage and palatoquadrate joint, (5) width between ceratohyal and palatoquadrate joint, (6) length of ceratohyals, (7) length of the ethmoid plate, and (8) length of Meckel's cartilage as imaged from the lateral side. The straight-line tool was used to measure most parameters, but for curved element the segmented-line tool was used. For mineralized tissues stained by alizarin red, nine elements were scored for either presence or absence of mineralization: (1) parasphenoid, (2) otoliths (all four present or not), (3) cleithrum, (4) notochordal sheath, (5) fifth ceratobranchial, (6) teeth (on fifth ceratobranchial), (7) opercles, (8) first branchiostegal ray, (9) entopterygoid. All studied parameters are depicted in Figs. 2A–2C.

## Gene expression analysis

RNA isolation was performed as described previously (*Gebuijs et al., 2020*). Individual larvae were transferred to 2-ml Eppendorf tubes containing a metal grinding ball and the total RNA of each sample was isolated. Firstly, the larvae were homogenized in 400 µl Trizol reagent (Invitrogen, Carlsbad, USA) using a grinding mill for 20 s at 20 Hz. Samples were incubated at room temperature for 5 min and 80 µl chloroform was then added. Tubes were shaken for 15 s, followed by incubation at room temperature for 2 min. Samples were centrifuged at 18,000 g for 10 min in a cooled centrifuge (4 °C), and 200 µl of the aqueous

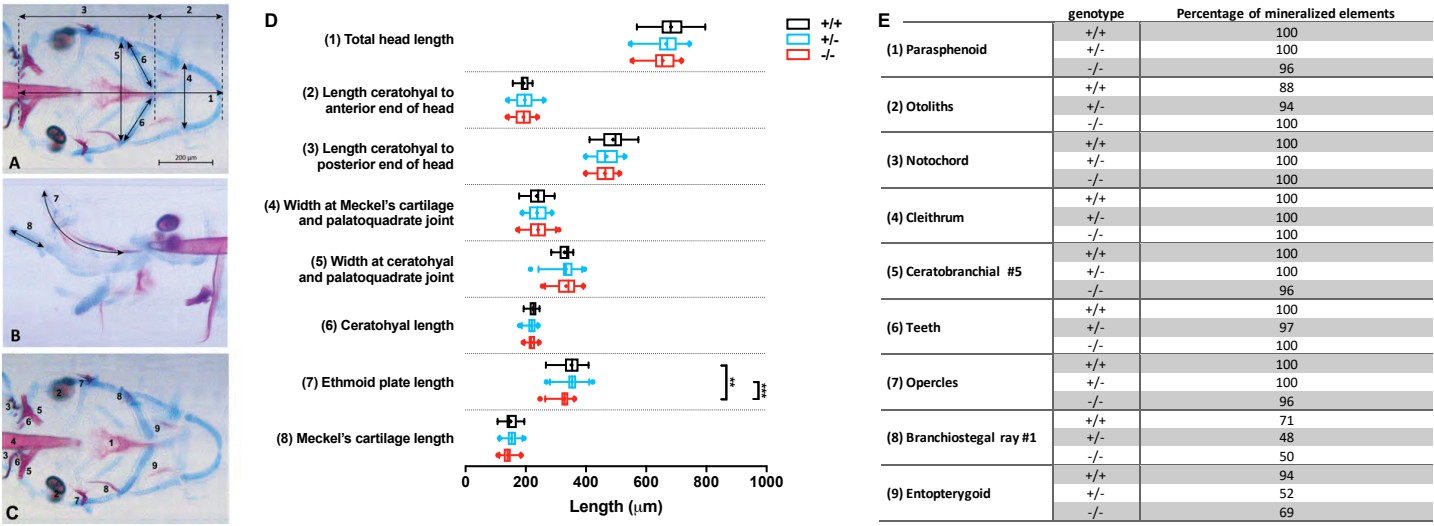

**Figure 2 Morphometrical analysis of cartilage and bone elements wild type, heterozygous and mutant *fgfr2* zebrafish at 5 dpf.** Morphometrical analysis of cartilage and bone elements wild type, heterozygous and mutant *fgfr2* zebrafish at 5 dpf. (A) Ventral view of wild type zebrafish stained for cartilage (blue) and bone (red) indicating the parameters 1–6 assessed in this study. The scale bar indicates 200 μm. (B) A lateral view to indicate morphometrical parameter 7 (the ethmoid plate) and 8 (Meckel's cartilage). (C) Ventral view to indicate the nine mineralized elements scored for presence or absence. (D) The length of the cartilage elements in the three genotypes of zebrafish: wildtypes (+/+, n = 17), heterozygotes (+/−, n = 28) and homozygous mutants (−/−, n = 26). The median is represented by the line within the box; the box shows interquartile range. The whiskers are drawn left to the 5th percentile and right to the 95th. Points beyond the 5–95 interval are drawn as individual dots. The mean is indicated by a "+" symbol. Data were statistically analyzed using non-parametric Kruskal–Wallis test and asterisks indicate statistical differences; **$p < 0.01$, *** $p < 0.001$. (E) Percentages of mineralized elements present in wildtypes (+/+, n = 17), heterozygotes (+/−, n = 31) and homozygous mutants (−/−, n = 26). Data were statistically analyzed by Fisher's exact test.

phase was transferred into a new tube. Isopropanol (200 μl) was added and mixed by inversion of the tube. The solution was stored at −20 °C for 1 h and then centrifuged for 15 min at 18,000 g in a cooled centrifuge. The pellet was washed with 75% ethanol, then air-dried for 10 min at room temperature and dissolved in 100 μl diethyl pyrocarbonate (DEPC)-treated water. Ten μl of a 3 M sodium acetate solution (pH = 5.4) and 250 μl 100% ethanol were added. Samples were stored overnight at −20 °C. The following day, the samples were centrifuged for 15 min at 18,000 g, the supernatant was decanted and the pellet was washed and then dissolved in 10 μl DEPC-treated water. The RNA concentration and purity of the samples were determined using a NanoDrop spectrophotometer at 260 nm wavelength (Thermo Fisher Scientific, Waltham, MA, USA).

The thus isolated RNA was treated with DNase to remove traces of genomic DNA. RNA (200 ng) was transferred into a PCR strip and DEPC-treated water was added to a total volume of 8 μl. Two μl DNase mix (1 μl 10X DNase I reaction buffer and 1 μl (1 U/μl) amplification grade DNAse I (both from Invitrogen, Carlsbad, USA)) was added and the solution was incubated for 15 min at room temperature. After incubation, 1 μl 25 mM EDTA was added to stop the DNAse reaction and the reaction mix was incubated for 10 min at 65 °C and stored on ice. Samples were used to synthesize cDNA by adding 1 μl random primers (250 ng/μl), 1 μl 10 mM dNTP mix, 4 μl 5X 1st strand buffer, 1 μl 0.1 M DTT, 1 μl RNase inhibitor (10 U/μl), 0.5 μl Superscript II (reverse transcriptase 200

**Table 1 Forward and reverse primers used for real-time qPCR for 17 developmental and two reference genes (*).**

| Gene | Forward primer sequence (5′→ 3′) | Reverse primer sequence (5′→ 3′) | Accession number |
|---|---|---|---|
| runx2a | TGTGGCTATGGCGTCTAACA | ATCTCCACCATGGTCCGGT | NM_212858.2 |
| runx2b | GGGCCAAACGCAGATTACAG | TCTGTCGAACCTGGAAGACG | NM_212862.2 |
| sp7 | GGATACGCCGCTGGGTCTA | TCCTGACAATTCGGGCAATC | NM_212863.2 |
| col1a1a | TTGCTTAGACCTGCGCTTCA | CCAGGGGGATTTTACACGCT | NM_199214.1 |
| col1a2 | GCTGGCCTTCATGCGTCTAC | ACACAGCCTTCTTCAGGTTTCC | NM_182968.2 |
| col2a1 | GCGACTTTCACCCCTTAGGA | TGCATACTGCTGGCCATCTT | NM_131292.1 |
| col9a1a | GGGGTGCGGTTGGATTTACT | TCCTGTCGATCCTTTCTCGC | NM_001130624.1 |
| col9a1b | AGACAAGTGTGCATGCGAGT | AAACCACTGTCACCTTGGGG | NM_213264.2 |
| dlx2a | GACTCAGTATCTGGCCTTGC | CTGCTCGGGTGGGATCTCT | NM_131311.2 |
| nkx3.2 | ACGCTAAAGCGCAAATCGAC | TTACAGTCGGACACGCAGTC | NM_178132.2 |
| fgf3 | GGTGGCAATCAAGGGACTGT | TGGTGCCGTGATGCATAAGT | NM_131291.1 |
| fgf8a | GCCGTAGACTAATCCGGACC | TTGTTGGCCAGAACTTGCAC | NM_131281.2 |
| fgfr1a | ATCTGTGTGATGGTGGGGAC | GATGAGCTGGAGTCCACAGAC | NM_152962.3 |
| fgfr1b | AGCATGGCTGACCGTTGTTA | GAGCTGGACTCCACTGACAC | NM_001161732.1 |
| fgfr2 | GCCTGTATGGTGGTGATCGT | TCTGACGACACTGTTACCTGG | NM_001243004.1 |
| fgfr3 | TCTCCGTGGTGAGGAAAGTC | TGCCATACAATGGGAAGGAGG | NM_131606.2 |
| fgfr4 | CTCTTGGACGTGTTGGAACG | CCACTGGATGTGAGGCTGAG | NM_131430.1 |
| elf1a* | CTGGAGGCCAGCTCAAACAT | TCAAGAAGAGTAGTACCGCTAGCATTAC | NM_131263.1 |
| rpl13* | TCTGGAGGACTGTAAGAGGTATGC | AGACGCACAATCTTGAGAGCAG | NM_212784 |

U/μl) (all obtained from Invitrogen) and 0.5 μl DEPC-treated water. The resulting mix was incubated for 10 min at 25 °C for annealing of the primers and then for 50 min at 42 °C for reverse transcription. Enzymes were inactivated by incubation at 70 °C for 15 min. Finally, the samples were diluted five times to serve as template in the qPCR reaction.

A real-time qPCR was carried out for each gene of interest. For each qPCR reaction, 4 μl of cDNA was mixed with 16 μl PCR mix (containing 10 μl SYBR green mix (2X) (BioRad, Hercules, CA, USA), 0.7 μl of each gene-specific primer (10 μM) and 4.6 μl water) was added to. Primers used are listed in Table 1. The qPCR reaction (3 min 95 °C, 40 cycles of 15 s at 95 °C and 1 min at 60 °C) was performed using a CFX 96 (BioRad, Hercules, CA, USA) qPCR machine. Threshold cycles (Ct values) were assessed and relative expression was calculated based on a normalization index of two reference genes: elongation factor alpha (*elf1a*) and ribosomal protein L13 (*rpl13*) (*Vandesompele et al., 2002*).

## Pharmacological inhibition of FGFRs

During two developmental time frames, 0–12 and 24–36 h post-fertilization respectively, FGFR1-4 were inhibited in wild type zebrafish using the pan-FGFR kinase inhibitor BGJ398 (NVP-BGJ398; Selleckchem, Houston, TX, USA). Four different final concentrations were used: 1, 2.5, 5 and 10 μM using dimethyl sulfoxide (DMSO) as vehicle. The final DMSO concentration was 0.1% (v/v) in all treatments. This concentration has been reported to be safe and not to cause mortality or malformations (*Hoyberghs et al., 2021*). Two control groups were included: one of 0.1% DMSO and an untreated group to check for possible

effects of the vehicle. After exposure, the media were removed and embryos were washed and placed back in fresh E3 medium.

## Genotyping

Genomic DNA of each individual stained larva was isolated and mixed with PCR mix and amplified using the following primers: fwd, 5′-AGCCAGTGCACAAGCTCACCGTA-5′ and rev, 5′-GTCCATGTAAACGTGCTTATTGTC-3′. The PCR products were digested with *RsaI*, loaded on a 2% agarose gel (SeaKem LE Agarose, Lonza) and separated by electrophoresis. The two possible digestion products are two wildtype fragments of 108 and 25 bp respectively, or a 133 bp mutant fragment.

Larvae to which qPCR was performed were genotyped on cDNA, using the following primers: fwd, 5′-AGCCAGTGCACAAGCTCACCGTA-5′ and rev, 5′-TCTGACGACACT GTTACCTGG-3′. The amplified DNA was digested with *RsaI* and separated on agarose gel by electrophoresis. The wildtype allele thus corresponds to fragments of 37 and 27 bp respectively; the mutant allele to a 64 bp fragment.

## Statistical analyses

All statistical analyses were performed using GraphPad Prism (version 5.03; GraphPad Software, La Jolla, CA, USA). If applicable, the data were first checked for normality with the D'Agostino-Pearson normality test. The data that were normally distributed were statistically compared using a one-way ANOVA followed by a post-hoc Tukey HSD test. Parameters that were not normally distributed were compared using the non-parametric Kruskal–Wallis test, followed by a post-hoc Dunn's multiple comparison test. Fisher's exact tests were performed to assess differences in presence of mineralized structures between wild type and mutant fish, or between controls and BGJ398 exposed fish. A threshold of $p < 0.05$ was considered significant for all statistical methods.

## RESULTS

### Subtle differences in cartilage and bone development in *fgfr2* mutants

Double staining of bone and cartilage was done in 26 homozygous $fgfr2^{sa10729}$ (−/−) mutants, 28 heterozygotes (+/−), and 17 wild types (+/+). We used eight morphometrical cartilage parameters and nine mineralized elements to assess the phenotypes of the craniofacial skeleton (Figs. 2A–2C). The −/− and +/− larvae showed hardly any significant differences in length of the eight measured cartilage elements when compared to wild types, as shown in Fig. 2D. The only exception was the length of the ethmoid plate, which was slightly shorter in the −/− compared to both the +/− and the wild type larvae. The marginal differences in the mineralized elements that were observed between wild types and mutants were all not significant (Fig. 2E).

### Subtle differences in gene expression in *fgfr2* mutants

To evaluate whether the mutation in *fgfr2* causes changes in the expression of genes involved in the FGF network, the relative expression of *fgfr1-4* was assessed, as well as that of 13 other genes related to bone and cartilage development (Fig. 3). Wild type siblings were

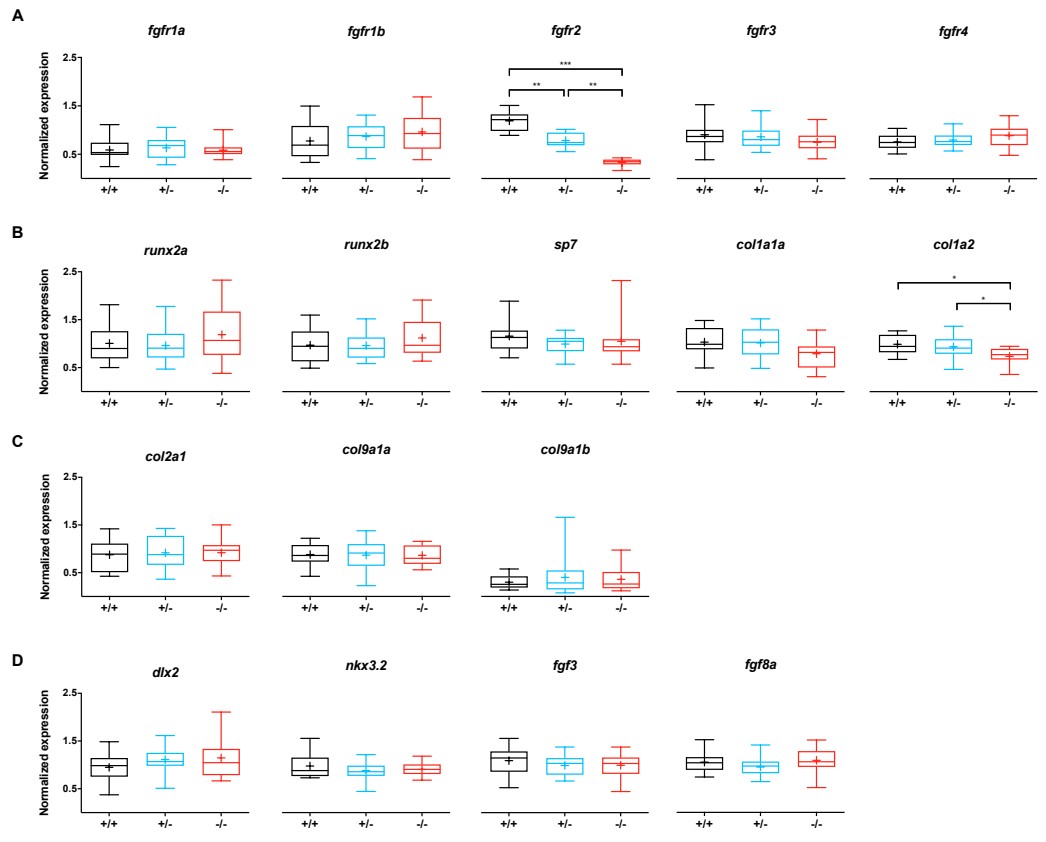

**Figure 3** **Relative expression levels of 17 genes at 5 dpf in wildtype ($n = 14$), _fgfr2_ $^{+/-}$ ($n = 18$) and $^{-/-}$ ($n = 12$) zebrafish larvae.** (A) Expression of _fgfr_ s reveals that _fgfr2_ is downregulated in our mutants. (B) Expression of genes involved in bone formation. (C) Genes involved in cartilage formation were not significantly different among groups. (D) Neural crest cell marker genes. In the box plots the median is represented by the line within the box; the box shows interquartile ranges. Whiskers are drawn down to the 5th and up to the 95th percentile. Points below and above the whiskers are drawn as individual dots. The mean is indicated by a "+" symbol. All the data were analyzed using a Kruskal–Wallis test followed by a Dunn's multiple comparison test. Asterisks indicate significant differences; *$p < 0.05$, **$p < 0.01$, ***$p < 0.001$.

compared to −/− mutant and +/− heterozygote _fgfr2_[sa10729] larvae at 5 dpf. The expression of _fgfr2_ was significantly affected in the +/- and -/- groups; compared to the wild types, the expression of _fgfr2_ was about 1.5-fold lower in +/− and about 3.5-fold lower in −/− larvae (Fig. 3A). The other receptor subtypes did not show differences in expression levels. For the 13 other genes, involved in bone formation (Fig. 3B), cartilage formation (Fig. 3C) and CNCC development (Fig. 3D), only _col1a2_ showed a significant downregulation in the −/− larvae compared to wild type. It was also significantly lower in −/− larvae compared to +/− larvae.
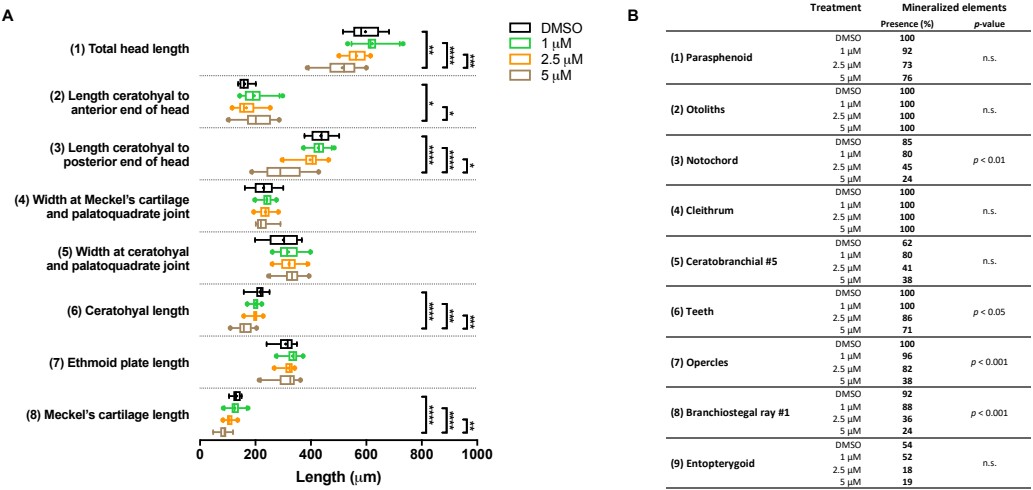

**Figure 4 Morphometric analyses of cartilage and bone elements in 5 dpf wild type larvae treated with BGJ398 during 0–12 hpf.** (A) The length of the cartilage elements. The controls were exposed to only the vehicle, DMSO at a final concentration of 0.1% ($n = 13$). The group sizes of the 1, 2.5 and 5 μM exposure were $n = 25$, 22 and 21, respectively. Medians are represented by lines within the boxes; the boxes show interquartile ranges. Whiskers are drawn left to the 5th percentile and right to the 95th. Points beyond the 5–95 interval are drawn as individual dots. The mean is indicated by a "+" symbol. Data were statistically analyzed by non-parametric Kruskal–Wallis test. Asterisks indicate significant differences; $^*p < 0.05$, $^{**}p < 0.01$, $^{***}p < 0.001$, $^{****}p < 0.0001$. (B) Percentages of mineralized elements present in DMSO controls ($n = 13$), 1 μM ($n = 25$), 2.5 μM ($n = 22$) or 5 μM ($n = 21$) BGJ398. Data were statistically analyzed by Fisher's exact test.

## Inhibition of FGFRs induces dose-dependent cartilage and bone malformations

As the *fgfr2* mutant larvae did not show a strikingly different phenotype compared to wild types, we investigated whether pharmacological blocking of FGFR1-4 affects craniofacial development. We exposed wildtype embryos to BGJ398, a potent and selective inhibitor of FGFR1-4 kinase activity, during two important developmental time frames: 0–12 and 24–36 h post-fertilization. These two windows correspond to the formation of the neural tube, and cranial neural crest cell differentiation, respectively. Fish were then allowed to further develop unexposed and skeletal staining was performed at 5 dpf.

In the groups that were exposed from 0 to 12 h after fertilization, the first observation was that all embryos died if exposed to the highest concentration (10 μM BGJ398). For the three lower concentrations, we observed a normal survival rate (>90%) but we found dose-dependent effects on some of the cartilage elements. In the 5 μM group, the length of five out of the eight analyzed cartilage elements was decreased (Fig. 4A). We also noted that in the 5 μM group in 52% of the larvae the Meckel's cartilage was underdeveloped or even absent. Differences in mineralized elements were observed for four of the nine analyzed elements, in the 2.5 as well as the 5 μM groups (Fig. 4B). The most marked effects were found for the notochord, the opercles and first branchiostegal ray.

In the groups treated with BGJ398 during 24–36 hpf, all larvae survived. For cartilage, we found significant size reductions in six of the eight elements analyzed, and the effects

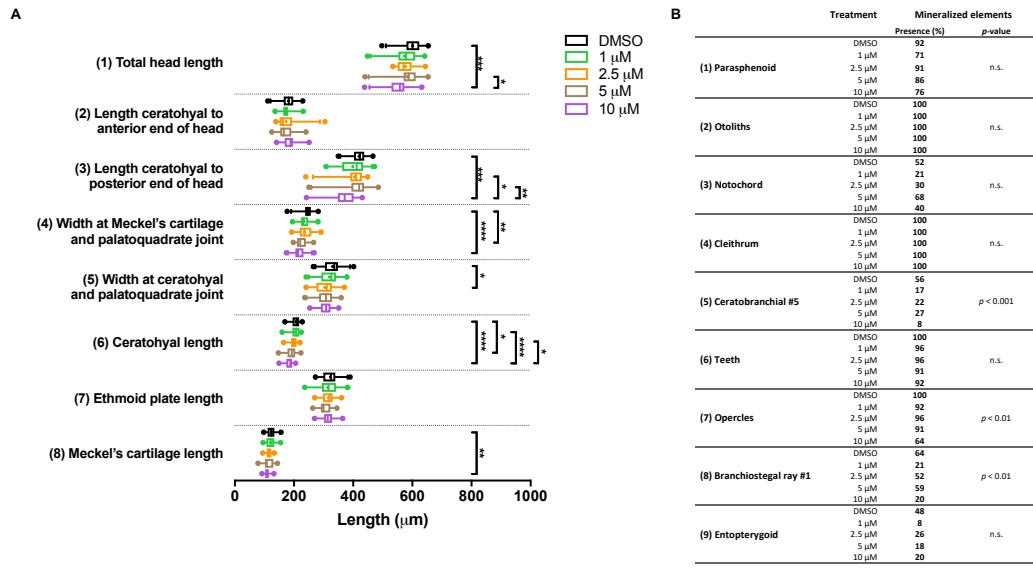

**Figure 5** Morphometric analyses of cartilage and bone elements in 5 dpf wild type larvae treated with BGJ398 dissolved in DMSO (final DMSO concentration 0.1%) during 24–36 hpf. (A) The length of the cartilage elements. The controls were exposed to only the vehicle, DMSO at a final concentration of 0.1% ($n = 25$). The group sizes of the 1, 2.5, 5 and 10 $\mu$M exposure were $n = 24, 23, 22$ and 25, respectively. Medians are represented by lines within the boxes; the boxes show interquartile ranges. Whiskers are drawn left to the 5th percentile and right to the 95th. Points beyond the 5–95 interval are drawn as individual dots. The mean is indicated by a "+" symbol. Data were statistically analyzed by non-parametric Kruskal–Wallis test. Asterisks indicate significant differences; $*p < 0.05$, $**p < 0.01$, $***p < 0.001$, $****p < 0.0001$. (B) Percentages of mineralized elements present in DMSO controls ($n = 25$), 1 $\mu$M ($n = 24$), 2.5 $\mu$M ($n = 23$), 5 $\mu$M ($n = 22$) or 10 $\mu$M ($n = 25$) BGJ398. Data were statistically analyzed by Fisher's exact test.

were dose-dependent (Fig. 5A). The largest differences in bony elements were observed in the 10 $\mu$M group (Fig. 5B) in which the mineralization of ceratobranchial 5 (but not the teeth on this element), the opercles and the first branchiostegal ray was often absent.

## Differential gene expression after 0–12 hpf and 24–36 hpf exposure to BGJ398

In the embryos treated with 5 $\mu$M BGJ398 during 0–12 hpf, gene expression levels of five *fgfr* and 12 other genes linked to cartilage and bone formation were measured at 5 dpf. Three of the five *fgfr* genes, namely *fgfr1a*, *fgfr1b* and *fgfr4* were upregulated compared to the DMSO control (Fig. 6A). Besides the receptors, also the osteoblast transcription factor *sp7* was upregulated (Fig. 6B). The genes *nkx3.2*, required for joint patterning and *dlx2a*, involved in the early specification and maintenance of CNCCs and later required for tooth development, were significantly upregulated (Fig. 6D). No down-regulated genes were observed.

In the embryos treated with 5 $\mu$M BGJ398 during 24–36 hpf we observed changes in expression of five of the 17 assessed genes. Four genes were downregulated, of which two were the receptor genes *fgfr1a* and *fgfr2* (Fig. 6A). Also, the bone matrix gene *col1a1a* and

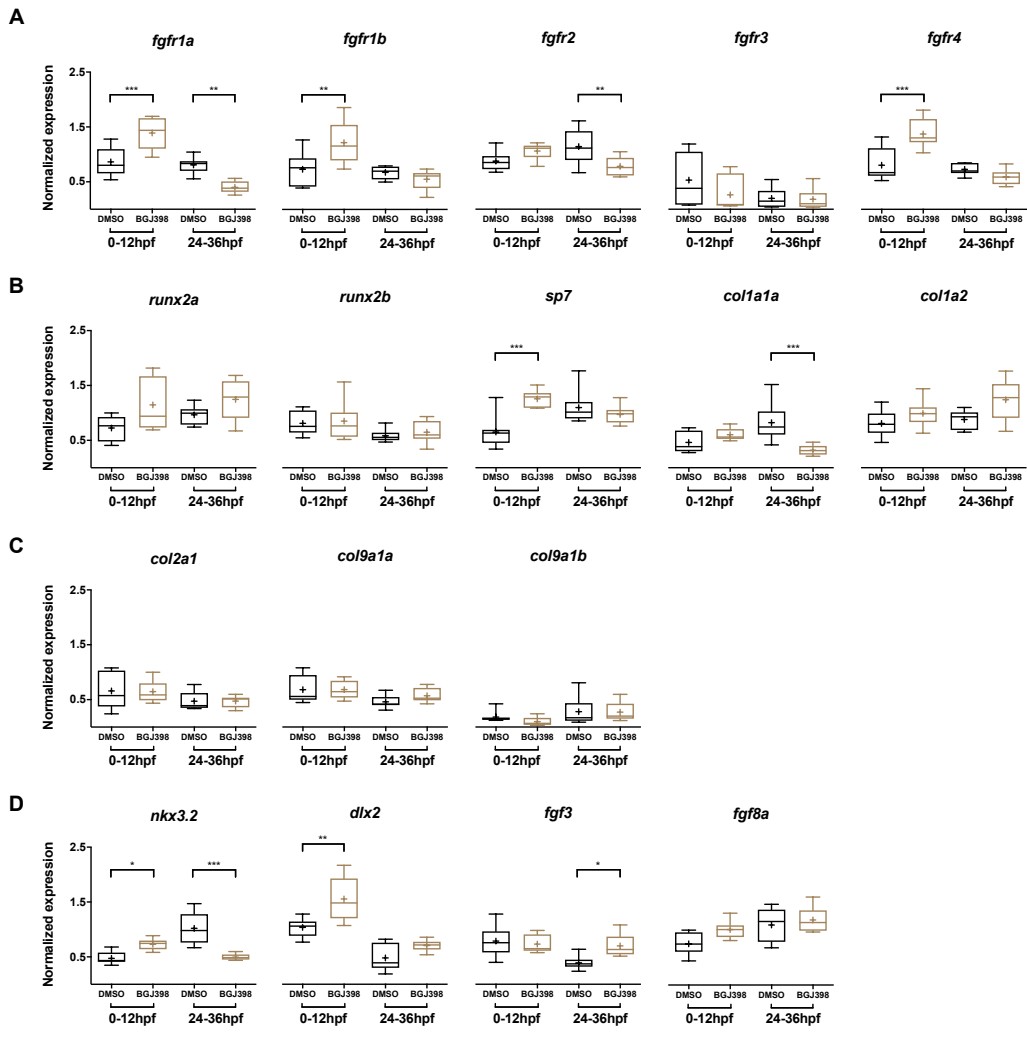

**Figure 6 Relative expression levels in 5 dpf wild type zebrafish larvae treated with 5 μM of BGJ398.**
Exposure windows were 0–12 and 24–36 hpf. Exposed fish ($n = 8$) were compared to a DMSO control ($n = 8$, final concentration 0.1%). (A) Expression of *fgfr* s showing differential expression in *fgfr1a*, *fgfr1b*, *fgfr2* and *fgfr4*. (B) Expression of genes involved in bone formation. (C) Genes involved in cartilage formation. From *col9a1b* one outlier ($p < 0.01$ in Grubbs' test) was removed. (D) Neural crest cell marker genes. All data are presented as box plots, with median represented by a line within the box; interquartile ranges by the boxes, 5–95 confidence intervals by whiskers and means indicated by "+" symbol. All the data were analyzed using a Kruskal–Wallis test followed by a Dunn's multiple comparison test. Asterisks indicate significant differences; *$p < 0.05$, **$p < 0.01$, ***$p < 0.001$.

*nkx3.2* were downregulated (Fig. 6B and 6D). The only upregulated gene was *fgf3* (Fig. 6D), while the remaining 12 genes did not show any significant differences in expression.

## DISCUSSION

Given the crucial role of FGF signaling in early skeletal development, we hypothesized that impairment of FGF signaling would affect the expression of genes related to bone and cartilage formation, leading to craniofacial malformations. Therefore, the aim of this study

was to assess the effects of an *fgfr2* mutation, as well as exposure to a general FGFR tyrosine kinase activity inhibitor on bone and cartilage development. We did not find an aberrant craniofacial phenotype in the *fgfr2* mutant zebrafish. By contrast, general kinase activity inhibition of all FGFRs induced severe defects in cartilage and bone formation, as well as a change in the expression of several developmental genes. From these findings we conclude that FGFR2 functions redundantly in the development of the craniofacial skeleton.

Our results show that the development of cartilage and mineralized elements in the larval head of homo- and heterozygous *fgfr2* mutant zebrafish is not significantly affected. In addition, the expression of genes required for cartilage and bone formation was also largely unaffected. These results are in contrast to an earlier study in which *fgfr2* was knocked down using antisense morpholinos; severe defects were shown in cartilage formation as well as a decrease in the expression of the cartilage-related genes *barx1* and *runx2b* (*Larbuisson et al., 2013*). The morpholino targeted the donor splice site in the first intron of the coding region, causing a deletion at the end of exon 2 resulting in a highly truncated and non-functional protein. In our study, the mutant has a point mutation after exon 9 and the resulting protein lacks the intracellular tyrosine kinase domain. Genetic compensation, an upregulation of compensatory genes that only occurs in mutants might be responsible for the difference in phenotype (*Rossi et al., 2015*). However, as the expression of the other *fgfrs* did not change in our mutant, genetic compensation seems unlikely to have occurred. The absence of an aberrant craniofacial phenotype in the *fgfr2* mutant may more likely be caused by a functional take-over by FGFR1, as has been reported in mammals (*Ornitz & Itoh, 2015*). Indeed, a study in zebrafish has shown FGFRs function redundantly; single ($fgfr2^{-/-}$) and double ($fgfr1a^{-/-};fgfr2^{-/-}$) mutant zebrafish did not show phenotypical differences and a triple ($fgfr1a^{-/-};fgfr1b^{-/-};fgfr2^{-/-}$) mutant was required to display severe craniofacial defects. The reported defects included missing ceratobranchials and ceratohyals, a misshapen palatoquadrate and a displaced Meckel's cartilage (*Leerberg, Hopton & Draper, 2019*). An intriguing finding in that same study was that all aberrant phenotypes required loss of function of FGFR1a, indicating a crucial role for this receptor in early development. The study further showed that only in the single and double mutants, mutant *fgfr2* mRNA was significantly decreased, suggesting that this mRNA is subject to nonsense-mediated decay. Expression levels of the other wild-type *fgfrs* were unchanged, indicating that also here genetic compensation does not seem to occur. These results are thus in agreement with our gene expression results in $fgfr2^{-/-}$. Altogether, these observations indicate that most of the mutated *fgfr2* mRNA is eliminated by nonsense-mediated decay and that other FGFRs can functionally compensate for this without any transcriptional changes.

The resulting truncation in the FGFR2 protein in our mutant is strikingly reminiscent of the FGFR5 protein, that contains three extracellular Ig-like domains and also lacks the intracellular protein tyrosine kinase domain (*Trueb et al., 2003*). Nevertheless, in mice it is still capable of signaling indirectly through dimerization with FGFR1 (*Amann & Trueb, 2013*). Moreover, it has been shown that the FGFR1, 2 and 3 can form heterodimers with each other (*Del Piccolo, Sarabipour & Hristova, 2017*). It is therefore intriguing to speculate

that the truncated FGFR2 protein in our zebrafish mutant heterodimerizes with other, unaffected receptors that are normally expressed and thereby can convey signals.

The lack of an aberrant craniofacial phenotype in the *fgfr2* mutant and the apparent redundancy of the FGFRs prompted us to block FGFR1-4 with the inhibitor BGJ398. It was applied during two developmental windows, 0–12 or 24–36 hpf. We chose these windows because in normal development, in the first 12 h the head and tail develop around the yolk sac (*Kimmel et al., 1995*) and at 10–11 hpf the neural tube is formed (*Araya et al., 2016*). From the neural tube neural crest cells, that contribute to cartilage formation in the head, originate and delaminate (*Kague et al., 2012*; *Schilling & Kimmel, 1994*). In the second window we chose, from 24–36 hpf, the cranial neural crest cells condensate in the oral ectoderm and differentiate into, among other cell types, chondroblasts that start forming the anterior neurocranium of the fish (*Kague et al., 2012*; *Wada et al., 2005*). Also other craniofacial bones, such as the opercles and branchiostegal rays, are neural crest cell derived (*Kague et al., 2012*). The exposure windows thus enabled us to study critically FGFR-dependent processes in craniofacial skeletogenesis.

We observed a high mortality in the group treated with the highest dose of BGJ398 during the first 12 h of development. Essential events such as cell proliferation and migration during neurulation and gastrulation, as well as brain development, patterning and morphogenesis occur in this phase, which are regulated by FGF(R)s (*Ota et al., 2010*; *Thisse & Thisse, 2005*). This could explain the high embryonic lethality that was observed. Treatment with lower concentrations of the inhibitor during 0–12 hpf resulted in dose-dependent differences in cartilage and bone elements in the zebrafish head at 5 dpf. The larvae treated with 5 µM BGJ398 showed a reduction in four cartilage parameters: the length of Meckel's cartilage, the ceratohyals and the (posterior part of the) head. These structures all originate from the pharyngeal arches in the zebrafish and are CNCC derived (*Kague et al., 2012*; *Schilling & Kimmel, 1994*). In the 0–12 hpf exposed group we also found that the Meckel's cartilage was severely affected or even entirely absent in about half of the larvae investigated. This may be caused by inhibition of FGFR3 in the zebrafish larval head. Earlier studies in chicken embryos that were depleted of FGFR3 also showed defects in Meckel's cartilage and reduced outgrowth of the mandibular bones (*Havens et al., 2008*). Also, mineralized elements that are (partly) of CNCC origin were often lacking, including the opercles, the branchiostegal rays and the teeth on ceratobranchial 5. Interestingly, the cleithrum, that does not originate from CNCCs (*Kague et al., 2012*), was not affected. These results point towards specific defects in craniofacial structures derived from CNCCs.

We observed an upregulation of *fgfr1a*, *fgfr1b* and *fgfr4* gene expression at 5 dpf in the fish treated with 5 µM BGJ398 from 0–12 hpf. These results confirm our hypothesis that these receptors are induced to compensate for the early inhibition by BGJ398. FGFR1 contributes to the development of teeth in zebrafish and mammals by induction of *dlx2* expression (62, 63). The latter gene has an important function in CNCC survival, but is also expressed during tooth development (*Borday-Birraux et al., 2006*; *Sperber et al., 2008*). Zebrafish embryos in which *dlx2a* was knocked down by morpholino injection showed increased apoptosis of CNCCs at 13 hpf, as well as malformed cartilage elements including Meckel's cartilage, ceratohyals and ethmoid plate at 5 dpf (*Sperber et al., 2008*).

These findings are thus consistent with the results from our own study and the early inhibition by BGJ398 may cause persistent expression of *dlx2a* and impairment of cartilage and tooth development. Also *sp7*, encoding a transcription factor crucial in osteoblast differentiation (*Li et al., 2009*), was upregulated at 5 dpf. As it thus controls the rate of bone growth, increased expression of *sp7* could be a compensatory reaction to the delayed bone formation.

In the larvae treated between 24–36 hpf we predominantly found significant differences for cartilage and bone elements in the group exposed to 10 μM. Like the 0–12 hpf group, Meckel's cartilage was smaller and the ceratohyals were notably shorter. Also the total head length was decreased. It has been reported that at 24 hpf, expression of *fgf3* in the posterior pharyngeal pouches is necessary for CNCC migration and differentiation (*David et al., 2002*). In that study, FGFRs were blocked with the synthetic inhibitor SU5402 for 24 h starting around 17 hpf and exposed larvae displayed severely reduced craniofacial cartilage structures and extensive cell death of the chondrogenic crest cells. Treatment with SU5402 from 15–39 hpf showed that reduced *fgf3* expression was the likely cause of reduced cartilage formation (*Walshe & Mason, 2003*). Consistent with our results, they found that the formation of ceratobranchial 5, bearing the pharyngeal teeth, was not affected. Therefore, we speculate that in our larvae there is a decreased number of CNCCs surviving and differentiating into chondrocytes. Accordingly, *fgf3* is upregulated at 5 dpf, which could also be a compensatory reaction for the inhibition earlier in development. In concurrence with the cartilage results, also dermal bones with a CNCC origin, the opercles and branchiostegal ray 1, were partially missing in the treated larvae.

## CONCLUSIONS

The zebrafish provides an attractive system to reveal how FGF signaling functions in early formation of the craniofacial skeleton. Mutations in specific protein domains of the FGFR2 are associated with craniofacial malformations in humans, including CLP. The high degree of conservation of FGF signaling between human and zebrafish warrants zebrafish in this type of studies. Our experiments show that the $fgfr2^{sa10729}$ point mutation does not severely affect craniofacial development in zebrafish, nor does it change the expression of genes required for bone and cartilage formation. This was unexpected, since a large part of the intracellular signaling domain of the protein is lost in this mutant. The lack of phenotype is presumably caused by redundancy between the FGFRs.

Pharmacological inhibition of all FGFRs during the earliest stage of development causes embryonic lethality or severe craniofacial malformations in structures derived from all pharyngeal arches. Treatment later in development induces malformations in some of the cartilaginous and bony elements. This shows that multiple FGFRs control craniofacial development in a coordinated way and compromised function affects CNCC survival (early exposure) or differentiation (late exposure). In addition, compensatory expression of genes essential for cartilage and bone formation occurs. We feel that these findings enhance our understanding of the etiology and genetics of craniofacial malformations. Moreover, the use of zebrafish as model system for genetic risk factors associated with cleft lip and palate

is increasingly acknowledged. We thus are convinced that the current study will broaden the insights into the etiology of this birth defect in humans.

### Funding

This study was supported by a research grant from the European Orthodontic Society and by a grant of the Dr. Vaillant Foundation. The funders had no role in study design, data collection and analysis, decision to publish, or preparation of the manuscript.

### Grant Disclosures

The following grant information was disclosed by the authors:
European Orthodontic Society.
Dr. Vaillant Foundation.

### Competing Interests

The authors declare there are no competing interests.

### Author Contributions

- Liesbeth Gebuijs performed the experiments, analyzed the data, prepared figures and/or tables, authored or reviewed drafts of the article, and approved the final draft.
- Frank A. Wagener conceived and designed the experiments, authored or reviewed drafts of the article, and approved the final draft.
- Jan Zethof performed the experiments, analyzed the data, prepared figures and/or tables, and approved the final draft.
- Carine E. Carels conceived and designed the experiments, authored or reviewed drafts of the article, and approved the final draft.
- Johannes W. Von den Hoff conceived and designed the experiments, authored or reviewed drafts of the article, and approved the final draft.
- Juriaan R. Metz conceived and designed the experiments, prepared figures and/or tables, authored or reviewed drafts of the article, and approved the final draft.

### Animal Ethics

The following information was supplied relating to ethical approvals (*i.e.*, approving body and any reference numbers):

NA; In the Netherlands experiments with zebrafish larvae up to 5 dpf do not require approval

### Data Availability

The morphometric data and the expression data are available in Supplementary Files.

### Supplemental Information

Supplemental information for this article can be found online at http://dx.doi.org/10.7717/peerj.14338#supplemental-information.

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
