# Peer review of "Targeting fibroblast growth factor receptors causes severe craniofacial malformations in zebrafish larvae"

_PeerJ, doi:10.7717/peerj.14338_

## Round 0.1 · original submission · Minor Revisions

The overall framework, structure, and flow of the manuscript are sound and well organized. However, on top of the reviewers' comments, here are some additional concerns. It would be helpful if the authors could provide more explanation on the detailed process of 'pharmacological inhibition of FGFRs', in particular, were the four concentrations of DMSO predetermined or how were they selected? Line 208: How long was the exposure duration?

Reviewer 1 ·

Basic reporting

This study focuses on the role of FGF in skeletal development. The main aim of this study is clear. Most of the parts in this manuscript are well-prepared and clearly written.

Experimental design

The experimental design answers the aim of the study. However, there are several issues that can be clarified:
- The experimental design should be clearly explained in brief in the methods section of the abstract.
- It is better to provide reference(s) for the statement in Line 133 – 134.

Validity of the findings

Most of the results are statistically- sound and well-presented.
I suggest adding the significance of this study to more significant perspectives or future perspectives in the conclusions.

Additional comments

Line 60 – 61: It is better to cite this sentence.
Please double-check all abbreviations used. Some were not appropriately introduced, e.g. Line 82: CLP, Line 95: SHH, etc.
I think the authors should add the significance of this study to more significant perspectives or future perspectives at the end of the introduction and conclusions.
Suggestion: Maybe the authors can add the number of samples used in the methods section even though it was mentioned in the results section.
Figure 1 consists of Figure 1A, 1B and 1C. There are no in-text citations for A, B and C, only Figure 1 in general.
It is better to enlarge the font used in the Figure 2 – Figure 6.
Line 263 – 267: these sentences sound like methods.

Reviewer 2 ·

Basic reporting

No comment

Experimental design

No comment.

Validity of the findings

no comments

Additional comments

The study has been well planned and executed. The write-up is all good and well elaborated. Hence no criticism from me.

---

## Round 0.2 · accepted · Accept

I thank the authors for following through on all the concerns and suggestions. The current version of the manuscript is acceptable for publication.